


# Rapid reduction of black carbon emissions from China: evidence from 2009–2019 observations on Fukue Island, Japan

Yugo Kanaya[1,2], Kazuyo Yamaji[2,1], Takuma Miyakawa[1], Fumikazu Taketani[1,2], Chunmao Zhu[1], Yongjoo Choi[1], Yuichi Komazaki[1], Kohei Ikeda[3], Yutaka Kondo[4], Zbigniew Klimont[5]

[1]Research Institute for Global Change, Japan Agency for Marine-Earth Science and Technology (JAMSTEC), Yokohama, 2360001, Japan
[2]Graduate School of Maritime Sciences, Kobe University, Kobe, 6580002, Japan
[3]Center for Global Environmental Research, National Institute for Environmental Studies, Tsukuba, 3058506, Japan
[4]National Institute of Polar Research, Tachikawa, 1908518, Japan
[5]International Institute for Applied Systems Analysis (IIASA), 2361 Laxenburg, Austria

*Correspondence to*: Yugo Kanaya (yugo@jamstec.go.jp)

**Abstract.** A long-term, robust observational record of atmospheric black carbon (BC) concentrations at Fukue Island for 2009–2019 was produced by unifying data from a continuous soot-monitoring system and a multi-angle absorption photometer. This record was then used to analyze emission trends from China. We identified a rapid reduction in BC concentrations of ($-5.8 \pm$
$1.5$)% y$^{-1}$ or $-48$% from 2010 to 2018. We concluded that an emission change of ($-5.3 \pm 0.7$)% y$^{-1}$, related to changes in China of as much as $-4.6$% y$^{-1}$, was the main underlying driver. This evaluation was made after correcting for the interannual meteorological variability, by using regional atmospheric chemistry model simulations (WRF/CMAQ) with constant emissions. This resolves current fundamental disagreements about the sign of the BC emission trend from China over the past decade, assessed from bottom-up emission inventories; our analysis supported inventories reflecting the governmental clean air actions
after 2010 (e.g., MEIC1.3, ECLIPSE v5a and v6b, and REAS updated) and recommended revision to those not (e.g., CEDS). Our estimated emission trends were fairly uniform over seasons but diverse among air-mass origins. Stronger BC reductions occurred in regions of South-Central East China, accompanied by CO emission reduction, while weaker BC reductions occurred in North-Central East China and Northeast China. Prior to 2017, the BC and CO emission trends were both unexpectedly positive in Northeast China during winter months, possibly influencing climate at higher latitudes. The pace of
emission reduction over China surpasses those of SSP1 scenarios (SSP: shared socioeconomic pathways) for 2015–2030, suggesting highly successful emission control policies. At Fukue Island, the BC fraction of PM$_{2.5}$ also steadily decreased over the last decade, suggesting that BC emission reduction started without significant delay with respect to other pollutants, such as NO$_x$ and SO$_2$, which are among key precursors of scattering PM$_{2.5}$.



# 1 Introduction

Atmospheric particles containing black carbon (BC) strongly absorb sunlight and reduce ice/snow surface albedo upon their deposition. Hence, their high emissions and atmospheric concentration levels in the present day, compared with those of preindustrial times, have contributed to global warming (IPCC, 2013). Their major sources are either anthropogenic fuel combustion or natural fires. Related to its rapid economic growth after 2000, China became the nation with the highest anthropogenic emission rate of BC, exceeding 1 Tg y$^{-1}$, which corresponds to ~30% of the world's total emissions (Bond et al., 2004; Zhang et al., 2009; Lamarque et al., 2010; Granier et al., 2011; Klimont et al., 2017; Li et al., 2017; Crippa et al., 2018). Therefore, its emission trend is of major concern to global air quality and climate change. Yet, fundamental disagreement remains among bottom-up emission inventories regarding the sign of the emission trend, especially for the period between 2005–2014, which represents the modern reference period for the Coupled Model Intercomparison Project Phase 6 (CMIP6) experiments. This is in a sharp contrast to our understanding of SO$_2$ and NO$_x$ emission trends from China (e.g., Zheng et al., 2018), for which studies have reached fairly consistent conclusions; reductions in their atmospheric concentration levels observed with satellites since 2008 or 2011 (Li et al., 2010, Krotkov et al., 2016) can be attributed to emission reduction, after the effects from meteorological variations and loss-rate variability were considered (e.g., Miyazaki et al., 2017). Here, the major difference between BC versus SO$_2$ and NO$_x$ is the unavailability of satellite observations that provide long-term, regional-scale information. Moreover, reliable, long-term mass concentration measurements at regionally representative ground-based sites useful for an emission trend analysis are scarce for BC. In the US and Europe, harmonized ground-based observations of BC (Murphy et al., 2011; Chen et al., 2012; Zanatta et al., 2016) are available and are used for estimations of emissions (e.g., Evangeliou et al., 2018). In Asia, Zhang et al. (2019) reported long-term trends in BC mass concentrations from China's BC observational NETwork (CBNET) covering urban and remote sites, measured using Aethalometers during 2006–2017. They found most of the stations had decreasing trends, ranging from 1.1 to 16.6% y$^{-1}$. However, there are issues with the interpretation of data from Aethalometers because of artefacts (Collaud Coen et al., 2010; Virkkula et al., 2007; Saturno et al., 2017).

In this context, long-term observations to determine BC emission trends in China are desired, based on reliable instruments with careful calibration. We have been conducting such observations of BC and related species at Fukue Island since 2009 (Kanaya et al., 2013, 2016; Miyakawa et al., 2017). The site receives air masses from various latitudinal bands of China during the winter monsoon (effective from autumn to spring), with negligible effects from local sources. It is unique that two instruments (COSMOS and MAAP) are regularly operated in parallel, their compatibility is assured, and that comparisons with other BC-measuring instruments have been successfully made on site, i.e., with a Single-Particle Soot Photometer (SP2) and an Elemental Carbon–Organic Carbon (ECOC) Analyzer (Miyakawa et al., 2017; Kanaya et al., 2013). As the major source region is located somewhat distant (ca. 700–1000 km) to Fukue Island, the variability in the wet loss term during transport, could be of concern. Nonetheless, previous analyses suggest that the accumulated precipitation along a trajectory (APT) is a useful index, which can be used to characterize the strength of the wet deposition (Oshima et al., 2012; Kondo et





al., 2016; Kanaya et al., 2016). Therefore, emission-specific characterization is possible, when data only with low APT (e.g., < 1mm for previous 72 h) are analyzed. In this paper, on the basis of this previously established approach, we provide an emission trend analysis of major Asian source regions, focusing on China, using the observation record of BC mass concentrations at Fukue Island for 2009–2019 (Sect. 3.1). The emission trend was analyzed, after removing the potential effects

of changes related to atmospheric transport (Sect. 3.2). The analysis includes discussion of how emission regions and seasonality affect the trend. In Sect. 3.3, we use this trend to evaluate bottom-up emission inventories for China for the past decade, as well as future shared socioeconomic pathway (SSP) scenarios. Furthermore, this same analysis was applied to emission trends of carbon monoxide (CO), a tracer for incomplete combustion in Sect. 3.4 to study the balance between BC and CO changes and to characterize the temporal evolution of the emission structure. Finally, in Sect. 3.5, the decreasing trend

in the BC fraction of $PM_{2.5}$ mass concentrations at Fukue is evaluated, showing that BC reduction has started without significant delay as $NO_x$ and $SO_2$, precursors of scattering or cooling aerosol particles.

## 2 Methodology

The Fukue Atmospheric Environment Observatory (32.75°N, 128.68°E) is located on Fukue Island (western Japan) within the East China Sea at an elevation of 75 m above sea level (asl) (Fig. 1). The main township of Fukue Island is located on its

southeast part, ensuring that the observatory in the northwest is free from local emission sources. Past observations at this site have shown that regional-scale characterizations of East Asian air pollution are tenable (Takami et al., 2005; Ikeda et al., 2014; Kanaya et al., 2016; Miyakawa et al., 2017).

Two filter-based BC-measuring instruments were regularly operated in parallel from April 2009 to May 2019 at this site. One is a continuous soot-monitoring system (COSMOS; BCM3130, Kanomax, Suita, Japan), and the other is a multi-angle

absorption photometer (MAAP5012; Thermo Scientific, Waltham, MA, USA). They are qualified such that the effects from co-existing scattering particles on BC observations are minimized. For the COSMOS, a pre-heater maintained at a temperature of 300°C removed the major, non-refractory fraction of scattering particles, and thereby the temporal change in filter transmittance related to BC was unambiguously measured. In addition, high-frequency measurements of the transmittance (at 1 kHz) enabled precise determination of this change (Miyazaki et al., 2008; Kondo et al., 2009, 2011a). For the MAAP,

measurements of reflectance at two angles with respect to the particle-laden filter were made, in addition to transmittance, and the built-in algorithm for radiative transfer enabled separation of scattering from absorption (Petzold and Schönlinner, 2004). We demonstrated that the two instruments had a high level of agreement and compatibility at Fukue, when the mass absorption cross-section for the MAAP was modified from manufacturer's recommendation (6.6) to 10.3 $m^2$ $g^{-1}$ (Kanaya et al., 2013, 2016). The filter-based observations were also successfully compared against an ECOC Analyzer (Sunset Laboratory, Inc.,

Tigard, OR, USA), where BC was calibrated as $CO_2$ after combustion, and against an SP2 instrument (Droplet Measurement Technologies, Inc., Longmont, CO, USA), where Fullerene soot (Alfa Aesar, USA) particles with known mass were used as a calibration standard (Miyakawa et al., 2017). The COSMOS and MAAP had a common inlet, equipped with a $PM_1$ size-cut



cyclone ($PM_{2.5}$ before 17 September 2011). Ohata et al. (2019) and references therein showed that the average accuracy of the COSMOS was estimated as ~10% under various field environments in the Arctic and in the East Asia. In this paper, the data records from the two instruments were 'unified' to minimize the gaps related to failure of individual instruments. An arithmetic mean was used to unify these data. When data from only one of the instruments were valid, data from the other instrument

were estimated using regression based on the nearest month for which data from both instruments were available. Averages of the slope and the correlation coefficients for monthly regression lines (COSMOS/MAAP) were 0.88 and 0.98 ($N = 107$). The uncertainty of the BC observations was estimated to be 12%.

Atmospheric CO mixing ratios and $PM_{2.5}$ mass concentrations were observed with a non-dispersive infrared sensor (Model 48C; Thermo Scientific) and with a hybrid system (SHARP5030; Thermo Scientific) based on nephelometry and β-ray

absorption measurements (Kanaya et al., 2016). Their uncertainties were estimated to be 4% and <14%, respectively. Although CO measurements were initiated in 2010, summertime data until June 2013 were unusable because of an artefact related to water vapor.

Backward trajectory analysis (NOAA HYSPLIT trajectory model; Stein et al., 2015; Stein and Stunder, 2017) was used to identify the air-mass origins and to extract hourly data without influence from wet deposition. A trajectory was drawn every

15    hour for the coordinates of the observation site at an altitude of 500 m asl, for a duration of 120 h. The meteorological field used was the Global Data Assimilation System (GDAS1). Similar to Kanaya et al. (2016), six air-mass origin regions (Regions I–VI) were identified (Northeast (NE) China, North-Central East China (N-CEC), South-Central East China (S-CEC), South (S) China, Korea and Japan) from the first border line to cross below 2500 m asl (Fig. 1). Some air masses, particularly those identified as being from Korea, were influenced from other regions (e.g., N-CEC). To characterize the emissions from Korea,

we extracted air masses that only passed over the Korean region (Region V'). Similar extractions were made for Japan (Region VI'), as some air masses traversed the continent beforehand. The APT was calculated as accumulated precipitation along trajectories integrated over the last 72 h (occurring mainly over the sea) and used as an index of the wet deposition effect (e.g., Oshima et al., 2012). Our previous analysis suggested that, on average, a 15-mm APT halves the BC concentration observed on Fukue Island (Kanaya et al, 2016). In this study, air masses without significant influence from wet deposition (<10% of BC

is lost) were extracted for analysis using the criterion of an APT of less than 1 mm.

The Weather Research and Forecasting (WRF) coupled with the Community Multiscale Air Quality (CMAQ) model with constant emission of BC and CO, was used to characterize effects of meteorology, particularly transport efficiency to Fukue Island. This WRF/CMAQ model included effects of stagnation and dilution/dispersion, in both horizontal and vertical directions. The details of the model setup have been described by Ikeda et al. (2014) and Yamaji et al. (2014). Briefly, the

WRF model (version 3.3.1; Skamarock et al., 2008), based on the National Centers for Environmental Prediction (NCEP) Final Analysis data (ds083.2), was coupled to the CMAQ (version 4.7.1; Byun and Schere, 2006) to simulate regional-scale air pollution with a horizontal resolution of 80 km and 37 vertical layers. The domain of the model is depicted in Fig. 1. The superior performance of the WRF model for the study region was confirmed in three dimensions (Li et al., 2013). The fifth generation CMAQ aerosol module (AERO5) was used for the BC simulation, while the Statewide Air Pollution Research



Center Version 99 (SAPRC99) was used for air chemistry. Because our analysis was made on selected hourly data showing insignificant effects from wet deposition, (i.e., having APT values of less than 1 mm), representation of wet deposition in the model was not important in this study. The boundary concentrations were derived from climatological means from the global chemical transport CHASER model (Sudo et al., 2002). Fixed anthropogenic emissions were from monthly data of the

5 Regional Emission inventory in Asia (REAS; version 2.1), with a 0.25°×0.25° resolution (Kurokawa et al., 2013) for 2008; hereafter referred to as REAS2.1 (2008). We adopted the Global Fire Emission Database (GFED; version 3.1) for monthly emissions from biomass burning. Note that BC from natural fires in China was estimated to be only several percent of that from anthropogenic emissions (Yin et al., 2019). The model simulations were performed from April 2009 to May 2019. Hourly outputs at the nearest grid were used for analysis. Separately for observations and simulations, hourly BC concentrations

satisfying the low APT criteria (i.e., < 1mm for previous 72 h) were annually averaged for all air masses or for individual air-mass origin areas. As the emission rates were annually unchanged in the model simulations, the model results represented interannual variations (IAVs) in the meteorology. Therefore the only factor that the model failed to replicate the observation was the emission trend. The residual ratio of the observed to modeled BC mass concentrations for all or individual regions (and for a particular year, $y$) embodies information for a correction factor for the emission rate with respect to REAS2.1(2008),

i.e., E($y$)/REAS2.1(2008). This factor can be used to bring the model into agreement with observations, assuming the interior geographical patterns of emissions for individual regions are unchanged from 2008 to the study period. Here, the function connecting emissions to observations was regarded as linear, because there were no known feedback processes linking wet deposition to the source term (emissions). The annual estimation of E($y$)/REAS2.1(2008) and its trend analysis were also done at a seasonal level.

## 3 Results and Discussion

### 3.1 Observed black carbon mass concentrations and trends

Monthly statistics for BC, CO, and PM$_{2.5}$ mass concentrations or mixing ratios and their trends for 2009–2019 are shown in Fig. 2. In the case of BC, the whole 122 months were covered by unifying data from the COSMOS and the MAAP. A long-

25 term decreasing trend was apparent, in which annual averages of BC mass concentrations decreased from 0.412 μg m$^{-3}$ in 2010 to 0.214 μg m$^{-3}$ in 2018, representing a 48% reduction. Superimposed on this downward trend, a seasonal trend with summer (June-July-August; JJA) minima, and some increases during early 2014 were evident. The cause of the summertime minima was the intrusion of clean air masses from the Pacific Ocean, as discussed previously (Kanaya et al., 2016). For both CO and PM$_{2.5}$, clear decreasing trends were detected, in which annual averages decreased from 206.6 ppbv in 2014 to 164.9

30 ppbv in 2018 for CO, and from 18.0 μg m$^{-3}$ in 2010 to 14.2 μg m$^{-3}$ in 2018 for PM$_{2.5}$.

The trends in spring (March-April-May; MAM) are shown in Fig. 3 (plus signs indicate maxima; error bars indicate 90th and 10th percentiles; boxes indicate 25th and 75th percentiles, with median values shown as horizontal bars and averages as circles).





Decreasing trends were evident for the high percentiles (75th and 90th), as well as for averages. Three-year running means were also shown for medians, 90th and 10th percentiles (gray solid and broken lines) to smooth out interannual variations (IAVs). Least-square linear fittings were made for annual averages, showing negative slopes. For unified BC data, the slope was $(-0.031 \pm 0.008)$ $(1\sigma)$ $\mu g\ m^{-3}\ y^{-1}$, suggesting that the negative slope was significant at a $3\sigma$ or higher confidence interval

$(P = 0.0031)$. By dividing the slopes by the mean mass concentrations for all available data during the 2009–2019 period $(0.421\ \mu g\ m^{-3}$; Table 1), the relative springtime trends were evaluated as $(-7.5 \pm 1.9)\%\ y^{-1}$, implying that the emission rates from important source regions have decreased. We will conclude it by removing effects of meteorological variability, which were quantified with model simulations in the following sections.

     Table 1 summarizes the mean values and trends in the observed BC mass concentrations for all seasons and within each

season. The percentage trends were derived by dividing the trend by the mean concentrations over the 2009–2019 period. The ranges of the percentage trends were negative for all cases (at 95% confidence level, $P \le 0.016$), ranging from −5.8 to −7.5% $y^{-1}$, except during summer. Up to this point, the observational data were not selected using the APT criterion. If the trends are different for the subset of cases with APT < 1 mm, then an influence from variable wet deposition strength is implied. However, this hypothesis was not supported, as trends were indifferent. The subset of data with APT < 1 mm had a range from −6.2 to

15    −7.6% $y^{-1}$ $(P \le 0.019$, except during summer)

     Weak trends were recognized for APT (Fig. S1), for all seasons during 2009–2018 with a mean value of 7.4 mm and a trend of $(-0.09 \pm 0.08)$ mm $y^{-1}$ or $(-1.2 \pm 1.1)\%\ y^{-1}$ $(P = 0.31)$. When the precipitation amount was translated to BC transport efficiency (TE) using the previously established relationship: $TE = \exp(-A1 \times APT^{A2})$, where $A1 = 0.109$ and $A2 = 0.684$ (Kanaya et al., 2016), a +0.3% $y^{-1}$ increase of BC mass concentrations could be expected, because of the reduced amount of

precipitation. Seasonally, the negative APT trend was significant in winter (December-January-February; DJF) and summer; yielding expected increases of +0.6% $y^{-1}$ and +3.2% $y^{-1}$ BC mass concentrations, respectively. This may explain why negative trends were weaker for observed BC levels for all APT values, compared with those for APT < 1mm. However, the trend was opposite and of smaller magnitude than the overall decreasing trend of −5.8% $y^{-1}$, suggesting that the wet deposition was a minor contributing factor.

Changes in large-scale flow patterns could also be a potential contributor to this trend in BC observations. However, this is unlikely, as the frequencies for various air-mass origin areas were almost unchanged during the study period (Fig. S2). The dominant source regions for Fukue Island were identified as S-CEC and N-CEC, for the periods of 2010–2013 and 2015–2018, respectively. These results were consistent with those of our previous study (Kanaya et al., 2016).

**3.2 Estimating emission trends after removal of meteorological effects**

**3.2.1 Model performance to capture short-term variations**

     The temporal variations of the observed and modeled BC and CO concentrations are shown in Fig. 4; these are based on March–April 2018 data, when the EMeRGe-Asia campaign was conducted, which included airborne observations from a


German aircraft HALO over East Asia (Andrés-Hernández et al., in preparation, 2019). The gray and light magenta lines from the WRF/CMAQ model captured elevated concentrations of both BC and CO quite well, demonstrating the high performance of the WRF/CMAQ and its ability to reproduce the phenomena of regional air pollution over East Asia. The air-mass origins according to backward trajectory modeling are superimposed on the figure. The light blue vertical bands in Fig. 4a represent

periods when APT was higher than 1 mm. In the periods without bands (i.e., APT < 1 mm), concentration peaks are linked to Region V (Korea) during 11–12 March 2018, Region II (N-CEC) during 24–26 March, Region III (S-CEC) during 27–28 March, Region VI (Japan) from 30 March to 2 April, and Region IV (S China) during 9–10 April. There is a tendency for BC concentrations to be overpredicted in air masses from China/Korea, and underpredicted in those from Japan. CO concentrations were also underestimated for cases from China/Korea. The observed-to-modeled mass concentration ratios were statistically

analyzed to estimate correction factors for the emission rate (i.e., $E(y)$/REAS2.1(2008)) and its trend.

**3.2.2 Effects from meteorological interannual variability derived from simulations: extraction of emission trend**

The left panels of Fig. 5 depict the footprints (total trajectory hourly points per 1×1 degree grid, below 2500 m asl) for all regions (I–VI), as well as for individual air-mass origin areas (Regions I, II, III, IV, V' (Korea only), and VI' (Japan only)) for all seasons during 2009–2019. In the middle columns, IAVs of mean BC concentrations (APT < 1mm) assigned to individual

origin regions are depicted for observations as well as simulations of the WRF/CMAQ model with fixed 2008 emissions. Data for 2009 and 2019 were eliminated from the analysis of all seasons, because the observation periods did not cover a full year. Generally, the observed IAVs were sometimes well reproduced by the model. This is expected as the model captures pollution events transported over high emission grids and effects of meteorology (i.e., dilution, stagnation, and vertical diffusion), as demonstrated in the data from the EMeRGe-Asia campaign in the previous subsection. Then the residual ratios of the observed

to modeled BC mass concentrations for individual air-mass origin areas were shown in the right panels of Fig. 5, as estimation of $E(y)$/REAS2.1(2008). The trends of this ratio (Table 2) are discussed below.

     It should be noted that the ratios (Fig. 5; Table 2), after applying a smoothing filter (3-y running mean), have relatively smaller IAVs (with correlation coefficients closer to −1, with respect to year) than the raw observations (Fig. 2, Table 1). Therefore, the emission trends could be more precisely quantified, after removal of meteorological effects. For all regions (I–VI), the

emission change was estimated to be $(-5.3 \pm 0.7)\%$ y$^{-1}$ on average ($P = 0.00048$). This value was similar for each season, with values of $(-5.7 \pm 0.5)$, $(-6.4 \pm 0.8)$, $(-6.9 \pm 1.8)$, and $(-6.7 \pm 1.0)\%$ y$^{-1}$ for winter, spring, summer, and autumn, respectively. All values were negative, with statistical significance ($P \leq 0.01$).

     Regionally, the trend was weaker for Japan, N-CEC, and NE China, with values of $(-3.3 \pm 1.1)$, $(-2.1 \pm 0.9)$, and $(-5.2 \pm 3.4)\%$ y$^{-1}$, respectively; but stronger for S-CEC, S China, and Korea, with values of $(-5.9 \pm 1.2)$, $(-6.3 \pm 2.9)$, and $(-8.4 \pm$

$0.6)\%$ y$^{-1}$, respectively, for all seasons. In Japan, the reduction pace of emissions may have slowed down, after the earlier initial rapid decrease that occurred in populated areas (e.g., Kondo et al., 2012; Yamagami et al., 2019). For N-CEC, the trends for winter and spring were similarly low, $(-1.8 \pm 1.5)$ and $(-1.9 \pm 1.5)\%$ y$^{-1}$, when the data number was large. For NE China,



the trend was $(-2.4 \pm 3.5)\%$ y$^{-1}$ for winter. This south–north contrast may reflect changes in emission structure. These values are compared with CO changes in Sect. 3.4.

Figure 6 shows our analysis for spring. This season had the longest record, from 2009 to 2019 (or from 2010 to 2018, after 3-y smoothing). Qualitatively, negative trends in BC were found for all regions. Weaker decreases were evident for N-CEC

(third row from top) and Japan (lowest row), although this trend was not linear for the whole period. The general trend was negative from 2010 to 2012, stable during 2012–2016, and became negative again during 2016–2018. This pattern was also found in the plots for  all seasons (Fig. 5).

It should be noted that the observation-to-model ratio, E($y$)/REAS2.1(2008), carries important information about the absolute emission rate (e.g., Kondo et al., 2011b). A ratio exceeding unity indicates that the emission rate needs to be increased by that

factor for the studied region/year for the WRF/CMAQ to reproduce the observed levels; while a ratio below unity implies that a reduction is needed. However, the uncertainties of both model simulations and observations may play an important role. For example, if the model had weak vertical diffusion/mixing coefficients and a tendency to overpredict the surface concentrations with reference to those at higher altitudes, then the correction factor determined from this ratio would underestimate emissions. Here, we estimated the absolute model uncertainties to be a maximum 16%, in simulating surface BC concentrations under

conditions without wet deposition, considering both horizontal and vertical inhomogeneities in the model and the spread of multi model simulations of CO over China (Kong et al., 2019). Using an observational uncertainty of 12%, overall uncertainty was estimated to be 20%. However, when there are only systematic uncertainties, the trend in the ratio is unaffected.

In this context, cases (year, region) having ratios exceeding 1.2 or lower than 0.8 were highlighted, as being noteworthy. As shown in Fig. 5, for the four subregions in China, the original ratios for 2011 (3-y average) spanning a range of 0.8–1.3 became

lower, yielding values of 0.6–0.8 in 2017 (3-y average), suggesting that the REAS2.1's original emission ratio for 2008 was reasonable in 2011 but required major revision for 2017. For Korea, the values in 2011 were relatively high, with a range of 1.3–1.6 in 2011, but concentrated around 0.8 in 2017, suggesting upshift/downshift in emissions are necessary for early/later study periods, respectively. Likewise, for Japan, the range in 2011 was very high (1.3–2.0), needing a significant upshift in the BC emissions. In 2017, the ratio also was still over unity (1.2–1.4), requiring a smaller but relevant upshift.

### 3.3 Estimated black carbon emission trend and rates from China: Comparisons with bottom-up inventories

In this section, total BC emissions for China and its decadal trend were estimated and compared with bottom-up inventories and future projections. Firstly, all Chinese provinces were assigned to the four Chinese subregions (i.e., regions I–IV) defined in this study (Fig. S3). Provinces with annual BC emission rates exceeding 0.05 Tg (with REAS2.1 (2008)) were assigned as

follows: Heilongjiang and Liaoning were assigned to NE China; Hebei, Nei-Mongol, Shandong, and Shanxi to N-CEC; Anhui, Henan, Hubei, Jiangsu, and Sichuan to S-CEC; and Guangdong, and Guangxi to S China. The regression lines for all seasons (based on 3-y averages) were used to calculate E($y$)/REAS2.1(2008) ratios for each region to be used as emission correction factors. These values were multiplied with the sum of the province-level emissions from REAS2.1(2008) to obtain the





emissions over a given year for that region. Finally, the total emissions over all four regions was calculated for each year. The uncertainty range was evaluated as the interval between maximum and minimum values of: (1) the 20% uncertainty range discussed above; (2) estimates using raw 3-y running means before regression; and (3) estimates using raw yearly means. Term (3) had the largest interval in 2014, while in all other cases, term (1) was the main contributing factor. Despite such uncertainty,

the decadal decrease in BC emissions was evident.

These emission estimates were compared with bottom-up inventories after 2005 for China in Fig. 7a. Notably, EDGAR4.3.2 (Crippa et al., 2018) data up to 2012 and CEDS data for CMIP6 (Hoesly et al., 2018) until 2014 showed monotonic increases, and clearly differed from the decreasing tendency found in this study. The industry sector mainly drove the increase in CEDS from 2010 to 2014 (Fig. S4). Likewise, REAS updated (Kurokawa et al., in prep., 2019) and MEIC1.3 (Zheng et al., 2018)

data also showed increases until 2011 and 2012, but then decreased till 2015 and 2017, respectively. The ECLIPSEv5a (Klimont et al., 2017) showed a decrease of $-3.1\%$ $y^{-1}$ during 2010–2015. Our results support the decreases in these data, but suggest that the onset of this decrease was earlier than 2012. This indicates that emission policies for particulates were effective, even during the 11[th] 5-y planning period (2006–2010), reducing BC emissions. It should be noted that ECLIPSEv6b (Klimont et al., in prep., 2019) predicted a decreasing trend starting from 2005–2010, related to emission reductions within the residential

sector; while decreases during 2010–2015 were mainly driven by emission reductions within the industrial sector (Fig. S4). The decreasing trend of Chinese BC emissions found in this study of $-4.6\%$ $y^{-1}$ was in good agreement with the value of $-4.7\%$ $y^{-1}$ from MEIC1.3 (Zheng et al., 2018). The different emission trends among inventories mainly reflect whether the governmental clean air actions after 2010, particularly those enacted in 2012 or later during the 12[th] 5-y planning period, were taken into account. These actions were considered in REAS updated, MEIC1.3, and ECLIPSEv5a (and v6b); while CEDS only

considered the effect of policies for CO, $NO_x$, $SO_2$ but not for BC. Instead, CEDS uses BC emission factors from the Speciated Pollutant Emission Wizard (SPEW) database (Bond et al, 2007) and the recent control concept was introduced but only up to 2010 (Hoesly et al., 2018). It should be noted that this decreasing trend has not previously been recognized, even in a recent national review paper of bottom-up inventories (Li et al., 2017). Clearly, in the next generation of emission inventories for CMIP7, the sharp decrease in the BC emissions from China, identified in this study, needs be addressed.

In terms of absolute values, the emission rate from CEDS was much higher (by a factor of ~2; 2–2.5 Tg $y^{-1}$) than our current estimate and well outside its estimated uncertainty range. Our previous work (Kanaya et al., 2016; Choi et al., 2019), based on an independent approach, in which we multiplied the estimated BC/CO emission ratio with the CO emission rate, also did not support such a high emission rate from China. The ECLIPSEv6b result best matched our absolute emission rate. The largest difference between ECLIPSEv6b and CEDS for 2010 and 2015 occurred with the industry sector, and then with the residential

sector (Fig. S4).

The emission reduction trend was converted to a relative scale, by normalizing it to the average value for 2012–2015 (Fig. 7b). Here we used a Monte-Carlo approach to evaluate the annual average and uncertainty intervals. A small increase in 2014 was recognized, related to larger uncertainty for that year. Nonetheless, a clear decrease over the decade was evident. This relative trend was compared against those from future emission scenarios, namely, SSP scenarios, starting from 2015 (Fig. 7b). The





emission projection data for China were taken from Gidden et al. (2019) and the SSP Public Database hosted by the IIASA Energy Program (https://tntcat.iiasa.ac.at/SspDb/dsd?Action=htmlpage&page=50). Up to 2018, the pace of reduction was stronger than the lowest envelope projected in future scenarios SSP1-19 and SSP1-26, having trends of −3.9 and −3.5% y$^{-1}$, respectively, for the period 2015–2030. These scenarios had the most significant emission reductions of greenhouse gases and

short-lived climate forcers, consistent with sustainable development. This analysis suggests that the countermeasures implemented in China have been successful at reducing BC emissions.

**3.4 Comparative trends in CO emissions**

CO is a useful tracer for BC emissions, because they are co-emitted from incomplete combustion. Figure 8 shows our analysis

results for ΔCO for winter months. Here, ΔCO was defined similarly to our previous study (Kanaya et al., 2016), as the enhancement from baseline, determined as 5$^{th}$ percentile over the moving 14-day time frame (see Fig. 4b). Thus, ΔCO represents regional enhancement in both observations and model simulations. Although the model does not include IAVs of the CO mixing ratios at lateral boundaries, a fair comparison with observations is possible. The IAVs of the removal of CO via its reaction with OH or secondary production of CO from hydrocarbons were not taken into account, because the focus for

this study was on relatively fresh peaks with magnitudes of 50–250 ppbv. The size of the dataset was larger than that for BC, because we did not preselect data having APT <1 mm. Therefore, we achieved better statistical convergence than for BC.

As for BC, the IAVs of ΔCO linked to meteorology were well reproduced by the WRF/CMAQ model. We used the residual trend to approximate the emission trend for CO. As expected, a negative emission trend was inferred (Table S1; Fig. 8). We found trends of (−5.2 ± 0.9)% y$^{-1}$ for all seasons and all regions for 2014–2017 ($P = 0.029$) and (−7.3 ± 0.6)% y$^{-1}$ for winter

and all regions for 2012–2018 ($P = 8.0 \times 10^{-5}$), respectively. They were similar to (−7.8 ± 1.2)% y$^{-1}$ and (−5.5 ± 0.7)% y$^{-1}$ estimated for BC during the same period. This suggests that BC and CO emission rates from China decreased in parallel over the past decade. In Region II (N-CEC), a slower CO reduction rate was identified (−3.7 ± 0.8)% y$^{-1}$, while large reductions occurred in Regions III (S-CEC) and IV (S China), with rates of (−6.9 ± 1.5) and (−9.6 ± 2.0)% y$^{-1}$, respectively (for all seasons). This pattern was quite comparable to the case of BC. In Korea, the reduction in CO (−2.5 ± 1.5)% y$^{-1}$ was relatively

small compared with that for BC (−9.3 ± 0.6)% y$^{-1}$. In Japan, the reduction (−3.8 ± 0.4)% y$^{-1}$ was similar to that of BC, (−4.4 ± 2.8)% y$^{-1}$.

The direction of BC and CO emission changes in winter were compared for the period 2012–2018 (Fig. 9). A parallel reduction in BC and CO emissions was noticed, based on the averages of air masses from all regions (colored squares). In Region I (NE China), except for 2018, the BC and CO emissions increased (colored triangles), with the BC/CO ratio remaining

almost constant, in contrast to other regions of China. Compared with other regions, this region may have undergone different economic development, delayed implementation of pollution controls, or shift or even increase of coal use in residential sector from urban to rural areas. In winter, the emission sources are unlikely to be forest fires, because the region would have had snow cover. Given that this region is nearer to the high latitudes/Arctic, the evolution of its BC emissions needs to be carefully





watched in future, to assess its impact on BC transport to higher latitudes and/or snow albedo change. Itahashi et al. (2019) recently found that $NO_x$ emission from NE China exhibited a later peak between 2013 and 2016, consistent to the tendency for BC. Region II (N-CEC) showed a CO decrease but only a very weak BC decrease, suggesting that the BC/CO emission ratio increased. Zheng et al. (2018) for MEIC1.3 suggested for the whole nation that control on industrial emission will result in

larger reduction in CO than BC; this type of change, towards improved energy efficiency, might be dominant in this region. In Region III (S-CEC), the reductions of BC and CO were similarly strong, suggesting effective pollution control measures in this region.

### 3.5 Tendency of atmospheric black carbon/PM₂.₅ mass concentration ratio

We also found that the BC reduction occurred at a faster pace than that of atmospheric $PM_{2.5}$ mass concentrations at Fukue
Island (Fig. 10). Similar IAVs were found for mass concentrations of BC and $PM_{2.5}$ (upper panel) related to meteorology. Again, using the $BC/PM_{2.5}$ ratio (lower panel), these meteorological IAVs were removed, as in previous analyses. The fraction of BC to $PM_{2.5}$ (0.0205 on average) showed a decrease of $(-2.6 \pm 0.7)\%$ $y^{-1}$ with $P = 0.013$ suggesting that $PM_{2.5}$ is 'brighter' (containing less black carbon) than a decade ago with a 95% confidence interval. This constant decrease would be difficult to achieve, if the BC emissions increased, while $SO_2$ and $NO_x$ (as precursors of non-BC fraction of $PM_{2.5}$) emissions decreased,
as CEDS data indicate. We also confirmed that the trend remained unchanged, when we selected data with APT < 1 mm. Therefore, we concluded that BC emission reduction occurred without significantly delay with respect to reductions of $NO_x$ and $SO_2$, although secondary production of organic aerosols from natural sources, if constant, may help explain part of the reduction in the $BC/PM_{2.5}$ ratio over time. Importantly, the balance or lag between $SO_2$ and BC reductions could have strong climate effects. Using satellite data inversion, emission reductions from China were estimated to be 24–33%, or $(2.4–3.3)\%$
$y^{-1}$ for $SO_2$ during 2007–2016 (Qu et al., 2019) and $(3.0–5.4)\%$ $y^{-1}$ for $NO_x$ during 2011–2016 (Itahashi et al., 2019). Our estimated BC emission reductions of 4.6% $y^{-1}$ were nearly equal to or even faster than these rates.

### 4 Conclusions

At Fukue Island, a rapidly decreasing decadal trend in the atmospheric BC mass concentrations of $(-5.8 \pm 1.5)\%$ $y^{-1}$ was detected in the period between 2009–2019. By choosing air masses unaffected by wet deposition for analysis and by
normalizing the observed annual averages with those from model simulations, we eliminated the influence of meteorology and focused on residual trends. We attributed these residual trends to emission changes in major source regions, particularly within four Chinese subregions, according to backward trajectory analysis. The overall emission reduction trend was estimated to be $(-5.3 \pm 0.7)\%$ $y^{-1}$ and was governed by the trend in S-CEC $(-5.9 \pm 1.2)\%$ $y^{-1}$, while N-CEC China and NE China showed weaker reductions. We estimated a $-4.6\%$ $y^{-1}$ BC emission reduction from China over the period 2009–2018; this supports
MEIC1.3, ECLIPSEv5a and v6b, and REAS updated inventories, which reflect emission reduction policies occurring after 2012, although our results suggest an earlier onset to these reductions. CEDS estimates used in CMIP6, characterized by a

large emission rate ($>2$ Tg y$^{-1}$) and an increasing trend (until 2014), were not supported by this analysis. This indicates a need for revision of emission inventories for CMIP7; we suggest the following key potential factors leading to the discrepancy; recent air pollution legislation and its implementation in industrial sector, recent trends in residential coal use, and estimates for the open burning of municipal waste. The reduction in Chinese BC emissions appears to be even faster than the decline

5    estimated in the deep mitigation SSP1 scenarios for 2015–2030, suggesting successful implementation of the BC reduction policies. The BC/PM$_{2.5}$ mass concentration ratio at Fukue Island showed a decreasing trend, implying that BC reductions followed the SO$_2$ and NO$_x$ controls; the latter species are among key contributors to secondary PM$_{2.5}$ components. A limitation of this study is that the analysis was conducted on a single site. Future studies should involve long-term observational data at multiple sites, when available.

## Data availability

The observational data set for BC is collectively available from https://ebcrpa.jamstec.go.jp/atmoscomp/obsdata/.

## Author contributions

YKa designed the study, conducted analyses, and wrote the manuscript. KY and KI optimized and conducted WRF/CMAQ model simulations. YKa, TM, FT, CZ, YKom, and YKon contributed to observations and instrumentation. YC contributed to
verification of the criteria of APT to extract cases with negligible wet deposition. ZK provided interpretation and discussion of the emission reduction.

## Competing interests

The authors declare that they have no conflict of interest.

## Acknowledgments

This research was supported by the Environment Research and Technology Development Fund (S-7, 2-1505, 2-1803) of the Ministry of the Environment, Japan, by the KAKENHI grant numbers 16H01770, 25220101, and 18H04143, and by the ArCS (Arctic Challenge for Sustainability) Project of the Ministry of Education, Culture, Sports, Science and Technology of Japan. We gratefully acknowledge assistance from Mr. Minoru Kubo and Drs. Tamio Takamura, Hitoshi Irie, and Jun-ichi Kurokawa. We thank Dr. Trudi Semeniuk from Edanz Group (https://en-author-services.edanzgroup.com/) for editing a draft of this
manuscript.



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





**Table 1.** Observed trends in atmospheric black carbon (BC) mass concentrations at Fukue Island for 2009–2019, showing cases without and with data filtering based on the Accumulated Precipitation along Trajectory (APT) criterion, for all seasons and in each season. The BC mass concentrations are those unified from a continuous soot-monitoring system (COSMOS), and a multi-angle absorption photometer (MAAP).

| | Mean ($\mu g\ m^{-3}$) | Trend ($\mu g\ m^{-3}\ y^{-1}$) | Trend (% $y^{-1}$) | R | P |
|---|---|---|---|---|---|
| BC (unified: COSMOS & MAAP) (all APT) | | | | | |
| All seasons | 0.321 | $-0.019 \pm 0.005$ | $-5.8 \pm 1.5$ | $-0.64$ | 0.0066 |
| DJF | 0.407 | $-0.028 \pm 0.009$ | $-6.8 \pm 2.3$ | $-0.92$ | 0.016 |
| MAM | 0.421 | $-0.031 \pm 0.008$ | $-7.5 \pm 1.9$ | $-0.86$ | 0.0031 |
| JJA | 0.189 | $-0.011 \pm 0.005$ | $-5.9 \pm 2.8$ | $-0.17$ | 0.070 |
| SON | 0.309 | $-0.022 \pm 0.005$ | $-7.0 \pm 1.7$ | $-0.85$ | 0.0034 |
| BC (unified: COSMOS & MAAP) (APT < 1mm) | | | | | |
| All seasons | 0.417 | $-0.027 \pm 0.007$ | $-6.5 \pm 1.6$ | $-0.83$ | 0.0056 |
| DJF | 0.478 | $-0.036 \pm 0.012$ | $-7.6 \pm 2.6$ | $-0.72$ | 0.019 |
| MAM | 0.502 | $-0.035 \pm 0.010$ | $-6.9 \pm 2.0$ | $-0.75$ | 0.0073 |
| JJA | 0.241 | $-0.015 \pm 0.011$ | $-6.2 \pm 4.7$ | $-0.42$ | 0.23 |
| SON | 0.386 | $-0.030 \pm 0.007$ | $-7.6 \pm 1.9$ | $-0.82$ | 0.0035 |

NB: Winter, DJF; Spring, MAM; Summer, JJA; Autumn, SON.

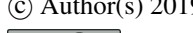



**Table 2.** Trends of estimated emission of black carbon (BC) for all or in each of the seasons, and for all regions (I–VI) and selected regions. The hourly data numbers ($N$), estimated emission ratios with respect to REAS2.1(2008), correlation coefficient regarding year ($R$), and $P$ values regarding significance of the trend are also listed.

| | $N$ | E(2011-17)/REAS2.1(2008) | Trend (y$^{-1}$) | Trend (% y$^{-1}$) | $R$ | $P$ |
|---|---|---|---|---|---|---|
| All seasons | | | | | | |
| All Regions I–VI | 31794 | 0.88 | −0.05 ± 0.01 | −5.3 ± 0.7 | −0.96 | 0.00048 |
| Region I (NE China) | 771 | 1.00 | −0.05 ± 0.03 | −5.2 ± 3.4 | −0.56 | 0.19 |
| Region II (N-CEC) | 2088 | 0.81 | −0.02 ± 0.01 | −2.1 ± 0.9 | −0.74 | 0.057 |
| Region III (S-CEC) | 2055 | 0.70 | −0.04 ± 0.01 | −5.9 ± 1.2 | −0.91 | 0.0043 |
| Region IV (S China) | 423 | 0.96 | −0.06 ± 0.03 | −6.3 ± 2.9 | −0.69 | 0.085 |
| Region V' (Korea only) | 4481 | 1.08 | −0.09 ± 0.01 | −8.4 ± 0.6 | −0.99 | 2.6 × 10⁻⁵ |
| Region VI' (Japan only) | 5217 | 1.36 | −0.04 ± 0.01 | −3.3 ± 1.1 | −0.80 | 0.030 |
| **DJF** | $N$ | E(2011-18)/REAS2.1(2008) | Trend (y$^{-1}$) | Trend (% y$^{-1}$) | $R$ | $P$ |
| All Regions I–VI | 12319 | 0.71 | −0.04 ± 0.00 | −5.7 ± 0.5 | −0.97 | 4.4×10⁻⁵ |
| Region I (NE China) | 124 | 0.84 | −0.02 ± 0.03 | −2.4 ± 3.5 | −0.27 | 0.52 |
| Region II (N-CEC) | 722 | 0.67 | −0.01 ± 0.01 | −1.8 ± 1.5 | −0.43 | 0.29 |
| Region III (S-CEC) | 787 | 0.59 | −0.03 ± 0.01 | −4.5 ± 1.5 | −0.78 | 0.024 |
| **MAM** | $N$ | E(2011-18)/REAS2.1(2008) | Trend (y$^{-1}$) | Trend (% y$^{-1}$) | $R$ | $P$ |
| All Regions I–VI | 11359 | 1.00 | −0.06 ± 0.01 | −6.4 ± 0.8 | −0.95 | 7.9×10⁻⁵ |
| Region II (N-CEC) | 1073 | 0.91 | −0.02 ± 0.01 | −1.9 ± 1.5 | −0.44 | 0.24 |
| Region III (S-CEC) | 1055 | 0.77 | −0.06 ± 0.01 | −8.1 ± 1.0 | −0.95 | 9.2×10⁻⁵ |
| Region V' (Korea only) | 1571 | 1.23 | −0.08 ± 0.01 | −6.5 ± 1.1 | −0.91 | 6.4×10⁻⁴ |
| Region VI' (Japan only) | 1667 | 1.39 | −0.06 ± 0.02 | −4.1 ± 1.3 | −0.78 | 0.014 |
| **JJA** | $N$ | E(2010-17)/REAS2.1(2008) | Trend (y$^{-1}$) | Trend (% y$^{-1}$) | $R$ | $P$ |
| All Regions I–VI | 3615 | 1.23 | −0.08 ± 0.02 | −6.9 ± 1.8 | −0.84 | 0.0088 |
| **SON** | $N$ | E(2010-17)/REAS2.1(2008) | Trend (y$^{-1}$) | Trend (% y$^{-1}$) | $R$ | $P$ |
| All Regions I−VI | 9373 | 1.05 | −0.07 ± 0.01 | −6.7 ± 1.0 | −0.94 | 0.00057 |
| Region I (NE China) | 277 | 1.12 | −0.11 ± 0.03 | −9.9 ± 3.1 | −0.79 | 0.019 |
| Region II (N-CEC) | 288 | 0.88 | −0.07 ± 0.02 | −8.3 ± 2.0 | −0.86 | 0.0057 |
| Region III (S-CEC) | 300 | 0.84 | −0.06 ± 0.01 | −6.6 ± 1.3 | −0.90 | 0.0025 |
| Region V' (Korea only) | 1721 | 1.21 | −0.07 ± 0.01 | −6.1 ± 1.0 | −0.93 | 0.00094 |
| Region VI' (Japan only) | 2369 | 1.42 | −0.08 ± 0.01 | −5.5 ± 1.0 | −0.92 | 0.0014 |

5   NB: N-CEC, North-Central East China; S-CEC, South-Central East China; NE China, Northeast China; S China, South China; Winter, DJF; Spring, MAM; Summer, JJA; Autumn, SON.



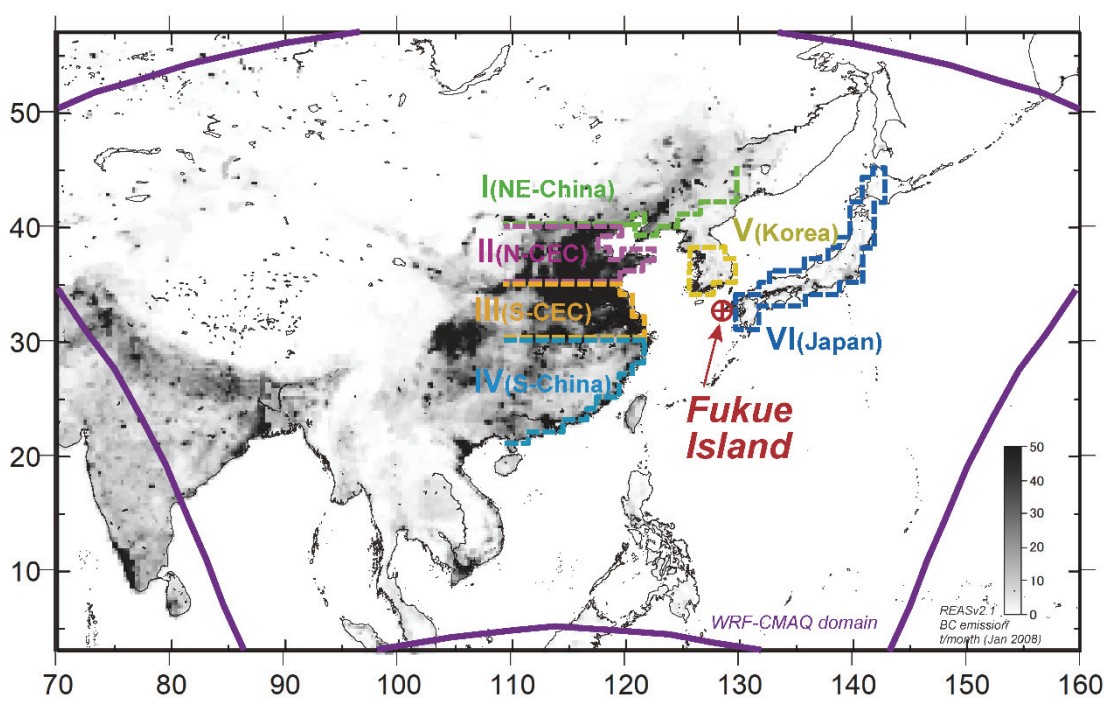

**Figure 1.** Location of Fukue Island (red), model domain (purple), and regional classification of air-mass origin areas (Regions I–VI). The background scale is black carbon (BC) emissions. N-CEC, North-Central East China; S-CEC, South-Central East China; NE China, Northeast China; S China, South China.



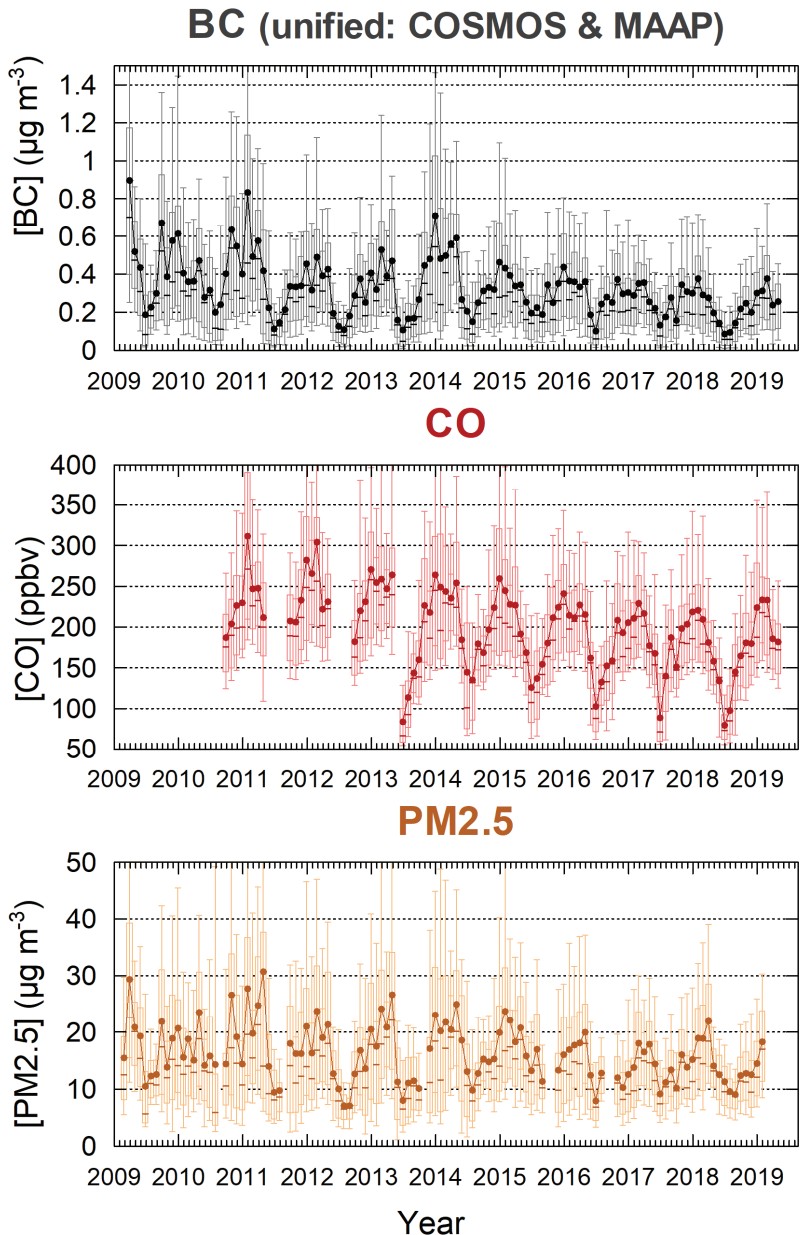

**Figure 2.** Monthly statistics of concentrations or mixing ratios of black carbon (BC), CO, and fine particulate matter (PM₂.₅) at Fukue Island during 2009–2019. Average (dots), interquartile range (open box), range from $10^{th}$ to $90^{th}$ percentile (error bars), and median (horizontal bars). BC measurements are from continuous soot-monitoring system (COSMOS) and a multi-angle absorption photometer (MAAP).

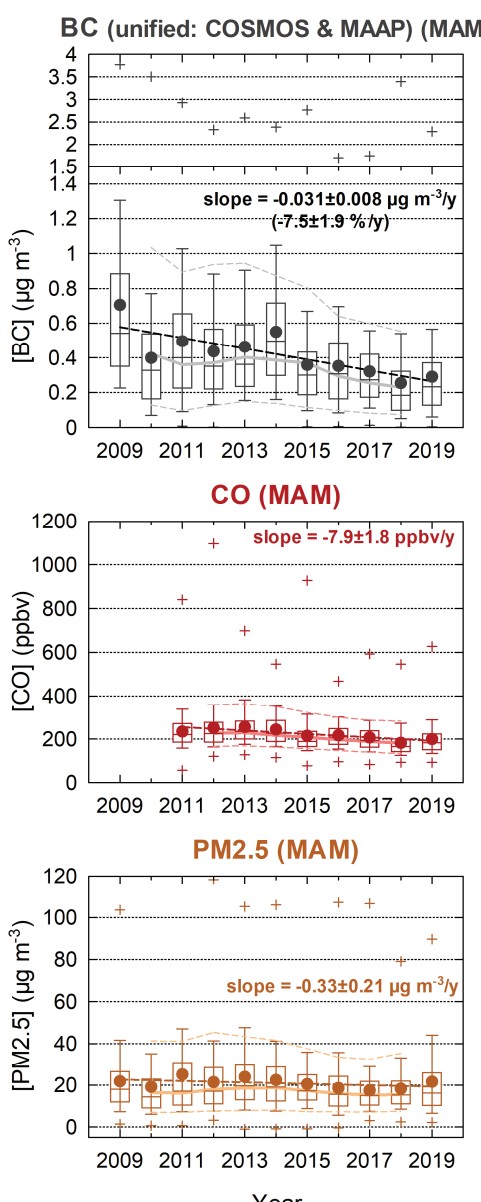

**Figure 3.** Statistics of observed black carbon (BC) mass concentrations, CO mixing ratio, and fine particulate matter (PM$_{2.5}$) mass concentrations during spring (MAM) from 2009 to 2019. Averages (dots), interquartile range (open boxes), 10$^{th}$ to 90$^{th}$ range (error bars), medians (horizontal bars), maximum and minimum hourly values (plus signs). Three-year running means from 2010 to 2018 for the medians, 10$^{th}$ and 90$^{th}$ percentiles are shown with connected solid/dotted lines. Regression lines for the three-year running means for the averages are shown as broken lines. BC measurements are from continuous soot-monitoring system (COSMOS) and a multi-angle absorption photometer (MAAP).





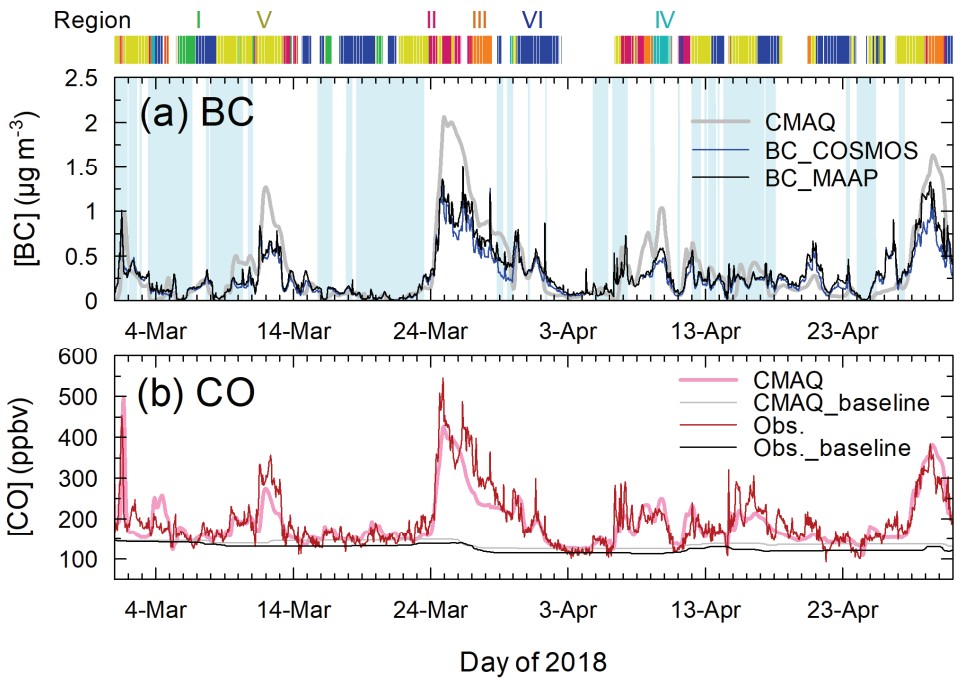

**Figure 4.** Black carbon (BC) mass concentrations and CO mixing ratios from observations and model simulations during the EMeRGe-Asia campaign. Top panel, the air-mass origin areas are indicated with colored vertical bars. The light blue bands in panel (a) indicate periods with Accumulated Precipitation along Trajectory (APT) > 1 mm.



**Figure 5. (Left)** Footprints, **(middle)** annually averaged black carbon (BC) concentrations for observations (blue dots) and model simulations (gray dots), and **(right)** their ratios used to estimate emission correction factors (E($y$)/REAS2.1(2008)) for all regions (I–VI) and individual regions (From top to bottom, as labelled in left), for all seasons. N-CEC, North-Central East China; S-CEC, South-Central East China; NE China, Northeast China; S China, South China.





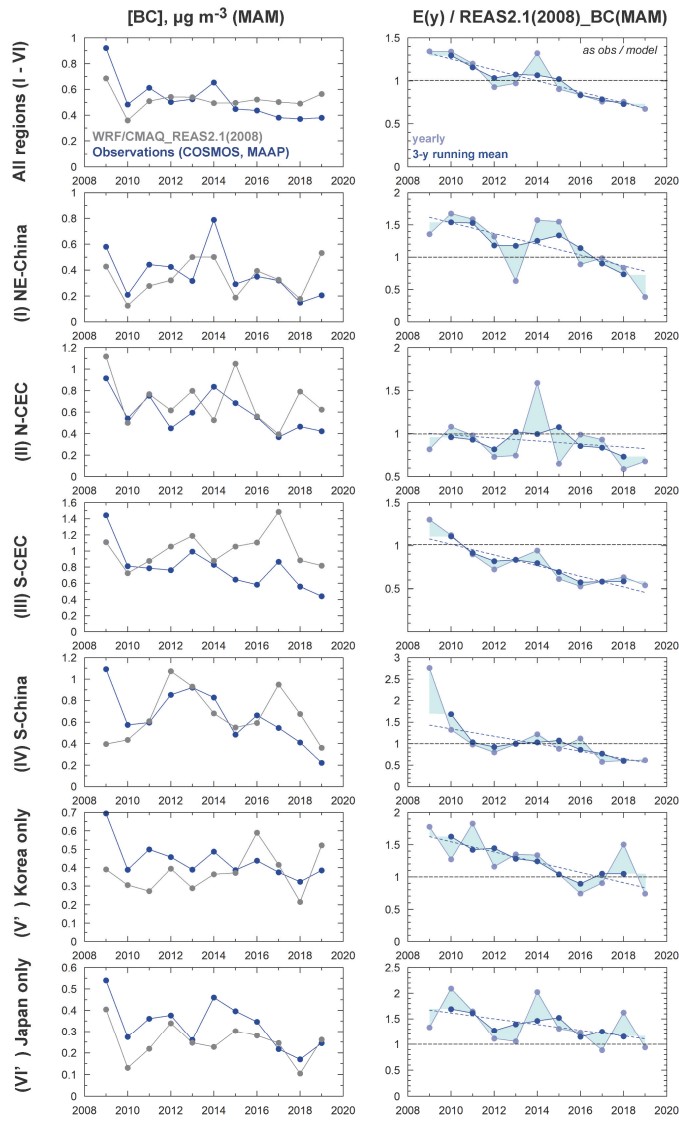

**Figure 6.** Same as Figure 5 but for spring (MAM), without footprint representation. Emission correction factor, (E(*y*)/REAS2.1(2008)); N-CEC, North-Central East China; S-CEC, South-Central East China; NE China, Northeast China; S China, South China.



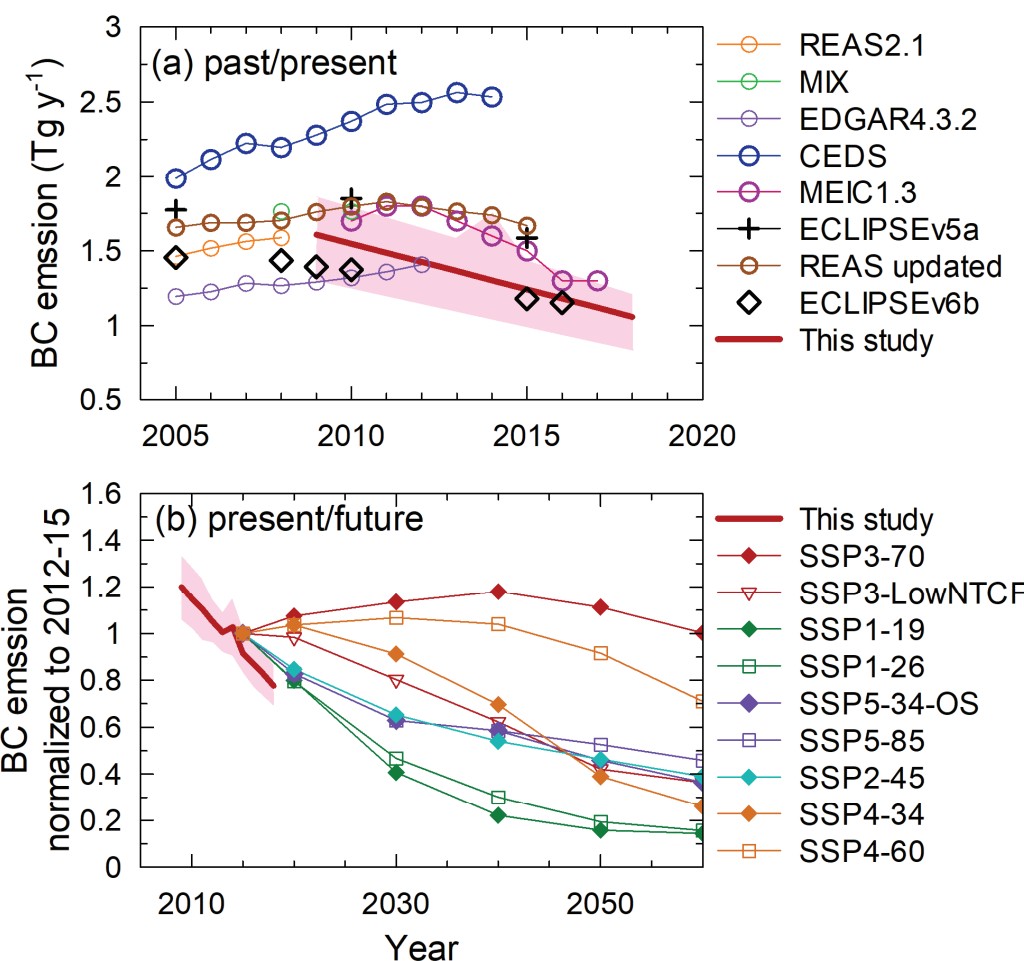

**Figure 7.** (a) Comparison of estimated black carbon (BC) emission trends (red closed circles) for China in past/present periods, compared with various bottom-up emission inventories shown in the figure legend. (b) similar plot showing comparisons with future emission scenarios shown in the figure legend, after normalization to 2012–2015. Shared socioeconomic pathway, SSP; near-term climate forcer, NTCF; overshoot scenario, OS.





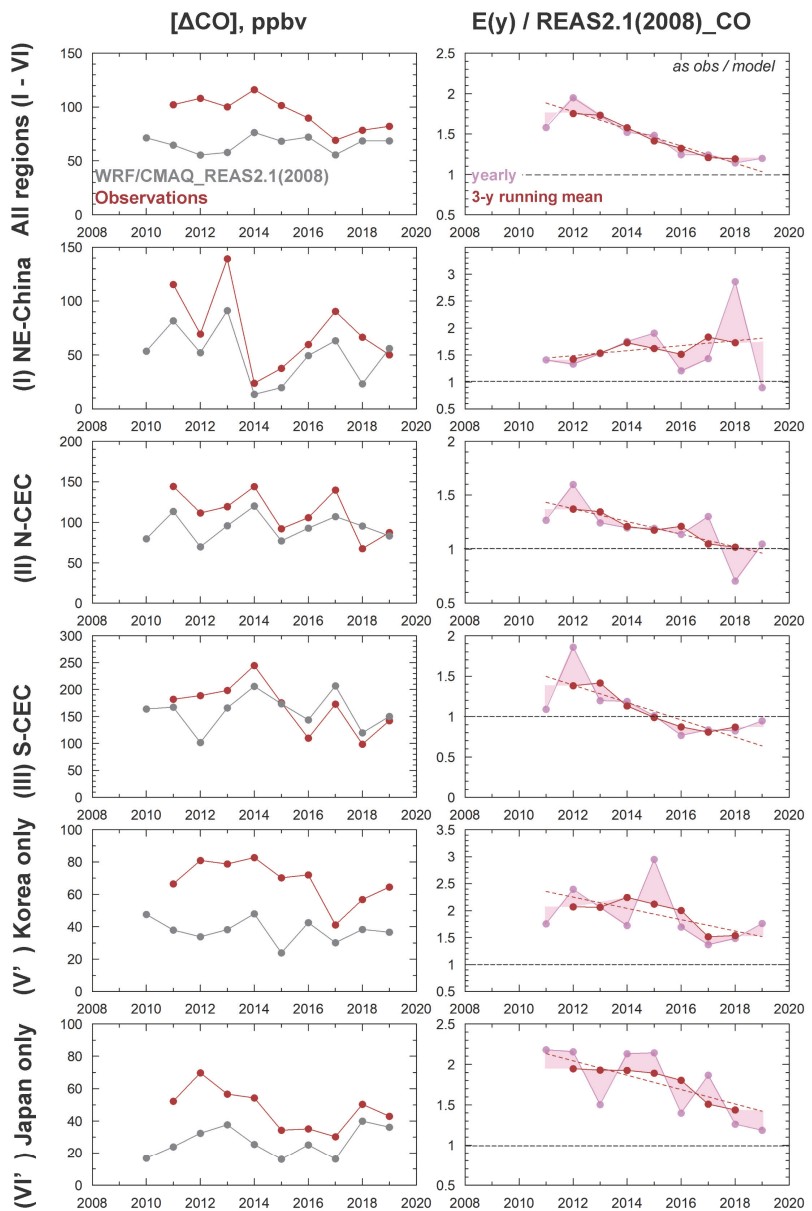

**Figure 8.** Same as Figure 6, but for ΔCO during winter (DJF). Emission correction factor, (E(*y*)/REAS2.1(2008)); N-CEC, North-Central East China; S-CEC, South-Central East China; NE China, Northeast China; S China, South China.





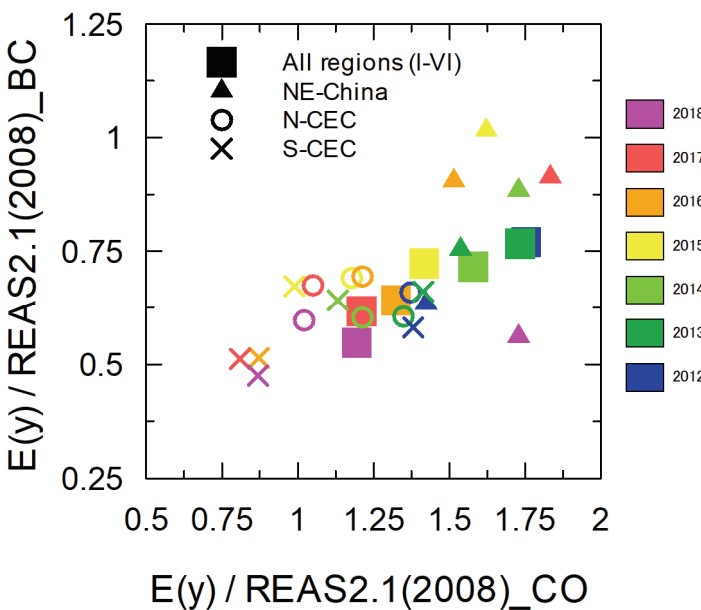

**Figure 9.** Relationship between trends in black carbon (BC) and CO emissions, as estimated from this study. Emission correction factor, (E($y$)/REAS2.1(2008)); N-CEC, North-Central East China; S-CEC, South-Central East China; NE China, Northeast China.





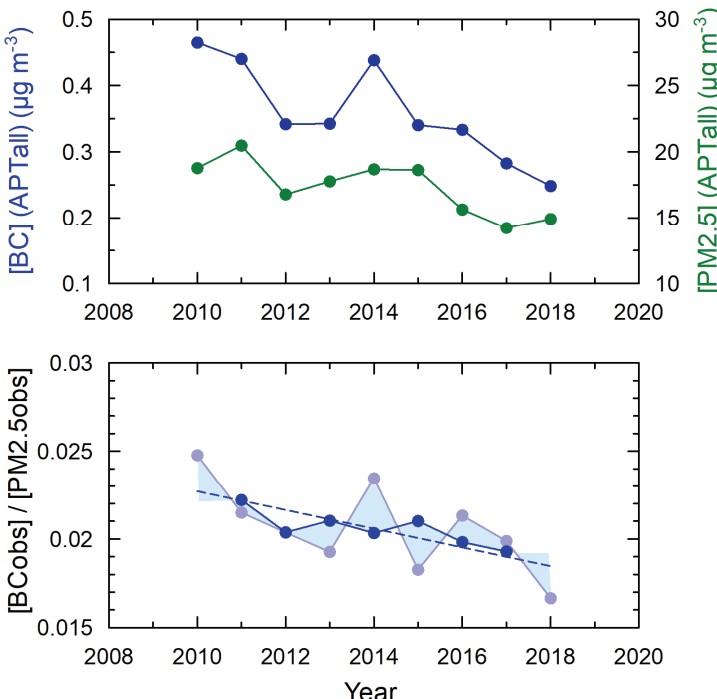

**Figure 10. (Top)** Temporal trends in the annually averaged black carbon (BC) and fine particulate matter (PM$_{2.5}$) mass concentrations for all cases. **(Bottom)** BC/PM$_{2.5}$ ratios showing a decreasing trend.

