# Peer review of "Rapid reduction of black carbon emissions from China: evidence from 2009–2019 observations on Fukue Island, Japan"

_Atmospheric Chemistry and Physics, 2019_

## Referee Comment (RC1) · Anonymous Referee #1 · 19 Jan 2020

The authors presented black carbon (BC) observations from 2009 to 2019 at the Fukue island located downwind of China and inferred China's BC emission trends from these observations. They used the chemical transport model WRF-CMAQ to estimate the meteorological effects on BC concentrations, used the backward trajectory model HYS-PLIT to attribute observed BC trends to emission source regions, and finally identified rapidly decreasing BC emissions from China, which are broadly consistent with the up-to-date bottom-up emission inventories. The comprehensive analysis that integrates several models and datasets gives a strong, convincing conclusion. This paper is well written and deserves publication after several issues addressed.

[Figure]

Major comments:

1) The key to this method is estimating emission correction factors, assuming that all of the differences between observed and modeled BC are attributed to the errors in surface BC emissions. This assumption has some problems since both models and observations have their own uncertainties. Without considering these uncertainties in the estimation of the emission correction factors, the authors tend to overestimate the uncertainties in BC emission inventories and thus tend to overestimate the correction factors. Although I believe that China's BC emissions have been declining since 2010, the authors need to justify the methodology they used and to acknowledge that the method still has large uncertainties in the emission correction factors. And if I understand correctly, the WRF-CMAQ model used here has no modules simulating BC wet deposition, which can cause <10% of BC loss even under an APT less than 1 mm. The current manuscript lacks a detailed discussion on the deficiencies of the model as well as the uncertainties in the observations.

2) In order to compare with bottom-up emission inventories, the authors scaled China's emissions from REAS2.1 with the emission correction factors for several large regions (Fig. S3). The footprint map of BC observations mainly covers central and eastern China (Fig. 5), while the emission correction factors are applied over the whole of China. I am wondering how many China's provinces and their BC emissions can be well observed by the station at Fukue Island. And the uncertainty range of the red curve in Fig. 7a should be larger than the current estimates after considering the representation errors of emission correction factors for the whole of China.

Minor comments:

1) Line 3 on Page 4. "minimize the gaps related to failure of individual instruments" Please clarify how many data points are missing from individual instruments. Are the BC trend estimates affected by the missing observation data?

2) Line 7 on Page 4. Please clarify why the uncertainty of the BC observations is

estimated at 12%.

3) Line 3 on Page 5. "representation of wet deposition in the model was not important in this study" I do not think so, because air masses without significant influence from wet deposition can still cause <10% of BC loss (Line 24 on Page 4).

4) Line 9 on Page 5. "Hourly outputs at the nearest grid were used for analysis" The nearest grid is the grid cell where the Fukue Island is located, right?

---

## Referee Comment (RC2) · Anonymous Referee #2 · 25 Jan 2020

This paper analyzed the declining trend in black carbon (BC) emissions from China, based on the long-term measurement data at a remote observation site in Japan. Combining air mass transport and air quality models, the authors made reasonable data filtering and simulation experiments. They drew a conclusion that China's BC emissions were clearly reduced in recent years, consistent with the big and continuous efforts of air pollution control by the government. In general the paper is well organized and written. Before it can be accepted for publication in Atmos Chem Phys, however, I have some concerns that should be more stressed or discussed. Some more detailed information should be provided as well, mainly in the measurement data and comparison between observation and modeling. The details follow.

[Figure]

1. The paragraph in Pages 3-4. The authors described the method of unifying the observations from COSMOS and MAAP, and stressed that the datasets correlate each other well. I suggest they provide the detailed correlation analysis between the two datasets with a figure, and indicated quantitatively the gaps between the two.

2. The third paragraph in Page 4. Does the author mean that the determination of influencing regions depend on the air mass transport modeling (HYSPLITT)? If so, I would suggest the authors provide the time-series or temporal variation of influencing regions within the research period, at least in the supplement. Some more discussions should also be given on the information.

3. Lines 11-16, Page 5. I cannot quite agree with the authors that the gap between observation and modeling indicates only the emission change, without the full evaluation of the model performance on 2008 (for which the emission data were applied). The determination of E(y)/REAS2.1(2008) thus seems problematic. How did the authors evaluate the model and recognize the modeling uncertainty for BC besides the wet deposition?

4. Figures 3 and 4. Why stress spring and select spring 2018 for comparing the modeling and observation results? Any special reasons?

5. Relevant with Question 3, I feel the authors need first to evaluate the model performance based on the observation, emission data and meteorology for the same year. The deviation between simulation and observation should be carefully studied to understand the uncertainty of modeling. Such bias should be excluded in the following step of determination of E(y)/REAS2.1(2008).

6. Small issue. What are the meanings of the dots with two colors in Figure 10b?

---

## Referee Comment (RC3) · Anonymous Referee #3 · 27 Jan 2020

The manuscript uses 11-year measurements of BC at a single site downwind of China to estimate emission trends of BC in different regions of China. Meteorological variability was estimated based on the WRF-CMAQ model with fixed emissions. Although only one site is used, the 11-year-long data record has valuable information that can be used to constrain source emissions.

My main concern is on the robustness of the regional trends derived with this single site. The regional trends were estimated by sampling observations and WRF-CMAQ model outputs by the footprint of HYSPLIT trajectories (c.f. Figure 5). Neither WRF-CMAQ model nor HYSPLIT footprint was evaluated with regards to their respective

ability to simulate interannual variability of meteorology, particularly precipitation. What is the resolution of the GDAS1 meteorology to drive HYSPLIT? The manuscript does not specify the resolution; it might be as coarse as 1degree x 1 degree. Since the authors used two models, they should be cross-validated. For validation of the HYSPLIT footprint, the footprint for one region should be compared to WRF-CMAQ simulation with that region's emissions turned off to validate if the two methods provide consistent variability of BC at the receptor site.

For the validation of WRF-CMAQ, although the manuscript did compare WRF-CMAQ model against the EMeRGe-Asia campaign, that comparison was only for one spring season (2018) while the main use of the model was to estimate interannual variations of BC. It is the WRF-CMAQ model's ability to simulate interannual variability of meteorology that should be evaluated. For example, the manuscript should present WRF-CMAQ simulated meteorological parameters at Fukue Island with those observed, and such comparison should be done for interannual scale. If data are available, the WRF-CMAQ model's meteorology over mainland China should be evaluated too.

---

## Author Comment (AC1) · 26 Mar 2020

**Response to the Reviewer #1**

The authors presented black carbon (BC) observations from 2009 to 2019 at the Fukue island located downwind of China and inferred China's BC emission trends from these observations. They used the chemical transport model WRF-CMAQ to estimate the meteorological effects on BC concentrations, used the backward trajectory model HYSPLIT to attribute observed BC trends to emission source regions, and finally identified rapidly decreasing BC emissions from China, which are broadly consistent with the up-to-date bottom-up emission inventories. The comprehensive analysis that integrates several models and datasets gives a strong, convincing conclusion. This paper is well written and deserves publication after several issues addressed.

We appreciate the reviewer's careful reading and positive comments on our manuscript. Detailed point-by-point responses are given below.

Major comments:
1) The key to this method is estimating emission correction factors, assuming that all of the differences between observed and modeled BC are attributed to the errors in surface BC emissions. This assumption has some problems since both models and observations have their own uncertainties. Without considering these uncertainties in the estimation of the emission correction factors, the authors tend to overestimate the uncertainties in BC emission inventories and thus tend to overestimate the correction factors. Although I believe that China's BC emissions have been declining since 2010, the authors need to justify the methodology they used and to acknowledge that the method still has large uncertainties in the emission correction factors. And if I understand correctly, the WRF-CMAQ model used here has no modules simulating BC wet deposition, which can cause <10% of BC loss even under an APT less than 1 mm. The current manuscript lacks a detailed discussion on the deficiencies of the model as well as the uncertainties in the observations.

We agree that the uncertainties of observations and models are important and will clarify this point in the revised manuscript:
"Therefore the only factor that the model failed to replicate the observation, except for their uncertainties, was the emission trend." (Page 5, Lines 33-34, in the track change document)

And the methodologies and results of the revised estimation of systematic and random uncertainties in the observations and models will be clarified:
" From the average and a standard deviation of the monthly MAAP/COSMOS ratios, systematic and random uncertainties were estimated as ± 14% and ± 17%, respectively (± 22% in total)." (Page 4, Lines 8-9)
" We estimated the model uncertainties from meteorology to be ~± 16%, in simulating surface BC concentrations under conditions with negligible wet deposition, considering both horizontal and vertical

inhomogeneities in the model and the spread of multi model simulations of CO over the East China Sea (Kong et al., 2019). The uncertainty was assumed to be contributed equally from systematic and random terms ($\pm$ 12 %)." (Page 5, Lines 25-28)

Then, the uncertainties in the absolute emission correction factors (E(y)/REAS2.1(2008)) and their trends will be clearly mentioned as follows:
"Overall uncertainty in the estimated E(y)/REAS2.1(2008) values was estimated to be $\pm$ 27%, including those random and systematic from both model and observation (see Sect. 2). On the other hand, the uncertainty in its trend was estimated to be $\pm$ 21%, as influenced only by random uncertainties." (Page 9, Lines 9-11)
In Fig. 7a, the uncertainty range (a band with pale red color) will be expanded accordingly, to cover the overall uncertainty.

Also, the fact that the WRF/CMAQ model included wet deposition will be clarified. The effect of the wet deposition on the used data set with the adopted low-APT criteria (<1 mm) in this study will be clarified, using the $\Delta BC/\Delta CO$ ratio (Kanaya et al., 2016), for both observations and model simulations:

" The median $\Delta BC/\Delta CO$ ratio for the observational data with APT < 1 mm ($N = 26423$) was only 1.9 % lower than that for data with APT = 0 mm ($N = 18907$), suggesting insignificant influence." (Page 4, Lines 33- Page 5, Line 2)
" Wet deposition was represented with the cloud_acm_ae5 module. Similarly to the observational data, the modeled $\Delta BC/\Delta CO$ ratio decreased with APT; the modeled median $\Delta BC/\Delta CO$ ratio for data with APT < 1 mm ($N = 26737$) was 3.7 % lower than that for data with APT = 0 mm ($N = 19197$). The removal in the model appeared stronger than the observational trend (1.9 %) but the error introduced due to the wet deposition representation was estimated to be small ($-2$ %) when using the adopted criteria (APT < 1 mm)." (Page 5, Lines 14-18)

2) In order to compare with bottom-up emission inventories, the authors scaled China's emissions from REAS2.1 with the emission correction factors for several large regions (Fig. S3). The footprint map of BC observations mainly covers central and eastern China (Fig. 5), while the emission correction factors are applied over the whole of China. I am wondering how many China's provinces and their BC emissions can be well observed by the station at Fukue Island. And the uncertainty range of the red curve in Fig. 7a should be larger than the current estimates after considering the representation errors of emission correction factors for the whole of China.

We agree that the low sensitivity to the emissions from inland areas increases the uncertainty. In the revised manuscript, after stating that total BC emissions for China and its decadal trend were estimated "on a best effort basis" (page 9, lines 26) in this study, a detailed analysis on this point will be provided as follows (page 10, lines 7-17):

One caveat in this analysis would be that the emissions from inland areas (e.g., Sichuan) produced less observational signal at Fukue than that from the CEC, while the footprint (Fig. 5) covered inland areas to 100° E well. We estimated 36% of BC emissions from China, particularly those from south or west areas including Sichuan, Guizhou, and Guangdong provinces (pale colors in Fig. S6), might have resulted in only 5% of signal at Fukue, when the pseudo signal synthesized by multiplying the footprint with the emission rates was analyzed. Thus any emission trends in such areas with less signal weight might be easily overlooked. A broken line in Fig. 7a represents a case when emission from such areas is assumed unchanged during the study period. Even in such a case, the trend was clearly negative and was within the range of overall uncertainty. We assumed that this scenario provided the possible weakest negative trend, considering that Zhang et al. (2019) reported negative trends in BC concentrations in those provinces similarly to CEC. Obviously reliable BC measurements at other locations with larger footprint over the areas are required to improve the analysis in the future. Here, nonetheless, we were able to conclude a decadal decreasing trend in the Chinese BC emissions from a long-term observation at a single site of Fukue.

Figure 7a and the supplementary figure (Figure S6 after revision) will be revised accordingly.

Minor comments:
1) Line 3 on Page 4. "minimize the gaps related to failure of individual instruments"
Please clarify how many data points are missing from individual instruments. Are the BC trend estimates affected by the missing observation data?

We will mention that the gap periods were 9673 and 10974 h (11 and 12 %) for COSMOS and MAAP, respectively (Page 3, Lines 4-5). The impact of the gaps on the overall trend was negligible.

2) Line 7 on Page 4. Please clarify why the uncertainty of the BC observations is estimated at 12%.

As mentioned earlier the statement on the revised estimation of the observational uncertainties will be revised as follows:
" From the average and a standard deviation of the monthly MAAP/COSMOS ratios, systematic and random uncertainties were estimated as ± 14% and ± 17%, respectively (± 22% in total)." (Page 4, Lines 8-9)

3) Line 3 on Page 5. "representation of wet deposition in the model was not important in this study" I do not think so, because air masses without significant influence from wet deposition can still cause <10% of BC loss (Line 24 on Page 4).

The effect of the wet deposition on the used data set with the adopted low-APT criteria (<1 mm) for this study will be clarified, using the ΔBC/ΔCO ratio (Kanaya et al., 2016), for both observations and model simulations:

" The median ΔBC/ΔCO ratio for the observational data with APT < 1 mm ($N = 26423$) was only 1.9 % lower than that for data with APT = 0 mm ($N = 18907$), suggesting insignificant influence." (Page 4, Lines 33- Page 5, Line 2)

" Wet deposition was represented with the cloud_acm_ae5 module. Similarly to the observational data, the modeled ΔBC/ΔCO ratio decreased with APT; the modeled median ΔBC/ΔCO ratio for data with APT < 1 mm ($N = 26737$) was 3.7 % lower than that for data with APT = 0 mm ($N = 19197$). The removal in the model appeared stronger than the observational trend (1.9 %) but the error introduced due to the wet deposition representation was estimated to be small ($-2$ %) when using the adopted criteria (APT < 1 mm)." (Page 5, Lines 14-18)

4) Line 9 on Page 5. "Hourly outputs at the nearest grid were used for analysis" The nearest grid is the grid cell where the Fukue Island is located, right?

Yes. We will state that hourly outputs at a grid including the Fukue site were used for analysis. (Page 5, Lines 29-30)

We thank the reviewer again for the productive comments.

**References**

Kanaya, Y., Pan, X., Miyakawa, T., Komazaki, Y., Taketani, F., Uno, I., and Kondo, Y.: Long-term observations of black carbon mass concentrations at Fukue Island, western Japan, during 2009–2015: constraining wet removal rates and emission strengths from East Asia, Atmos. Chem. Phys., 16, 10689–10705, https://doi.org/10.5194/acp-16-10689-2016, 2016.

---

## Author Comment (AC2) · 26 Mar 2020

**Response to the Reviewer #2:**

This paper analyzed the declining trend in black carbon (BC) emissions from China, based on the long-term measurement data at a remote observation site in Japan. Combining air mass transport and air quality models, the authors made reasonable data filtering and simulation experiments. They drew a conclusion that China's BC emissions were clearly reduced in recent years, consistent with the big and continuous efforts of air pollution control by the government. In general the paper is well organized and written. Before it can be accepted for publication in Atmos Chem Phys, however, I have some concerns that should be more stressed or discussed. Some more detailed information should be provided as well, mainly in the measurement data and comparison between observation and modeling. The details follow.

We thank the reviewer very much for reading our paper carefully and giving us valuable comments. Detailed responses to the comments are given below.

1. The paragraph in Pages 3-4. The authors described the method of unifying the observations from COSMOS and MAAP, and stressed that the datasets correlate each other well. I suggest they provide the detailed correlation analysis between the two datasets with a figure, and indicated quantitatively the gaps between the two.

In the revised supplementary material, time series plots during the whole period (2009–2019) and a correlation plot between the two data sets (i.e., COSMOS and MAAP) will be provided as Figures S1 and S2. In relation, the statements on the uncertainty estimation for the BC observations will be revised:
" From the average and a standard deviation of the monthly MAAP/COSMOS ratios, systematic and random uncertainties were estimated as ± 14% and ± 17%, respectively (± 22% in total)." (Page 4, Lines 8-9 in the track change document)

2. The third paragraph in Page 4. Does the author mean that the determination of influencing regions depend on the air mass transport modeling (HYSPLITT)? If so, I would suggest the authors provide the time-series or temporal variation of influencing regions within the research period, at least in the supplement. Some more discussions should also be given on the information.

Yes, we meant that HYSPLIT-based backward trajectories were used for the judgement of the influencing regions. In Fig. S1 of the revised manuscript, the assignment of the influencing regions during the whole study period will be shown. Figure S5a (formerly Figure S2a) showed statistics and trends of number of hourly cases of observed air masses from different origin areas. In text, we mentioned as follows (Page 7, Lines 17-18):
  Changes in large-scale flow patterns could also be a potential contributor to this trend in BC observations. However, this is unlikely, as the frequencies for various air-mass origin areas were almost unchanged during the study period (Fig. S5).

3. Lines 11-16, Page 5. I cannot quite agree with the authors that the gap between observation and modeling indicates only the emission change, without the full evaluation of the model performance on 2008 (for which the emission data were applied). The determination of E(y)/REAS2.1(2008) thus seems problematic. How did the authors evaluate the model and recognize the modeling uncertainty for BC besides the wet deposition?

5. Relevant with Question 3, I feel the authors need first to evaluate the model performance based on the observation, emission data and meteorology for the same year. The deviation between simulation and observation should be carefully studied to understand the uncertainty of modeling. Such bias should be excluded in the following step of determination of E(y)/REAS2.1(2008).

First, it should be noted that REAS2.1(2008) emission flux values were used as working references, and they were not necessarily true values for 2008 (Page 6, Lines 5-6 in the revised manuscript, track-change version); the true emission flux for 2008 is very likely different from REAS2.1(2008) and therefore difficult to perform the "evaluation" that the reviewer suggested. On the other hand, the superior performance of the WRF/CMAQ model incorporating REAS2.1(2008) emission in simulating peaks and relative variations is demonstrated in Fig. 4 (for spring 2018). Similarly good performance will be shown for the other periods (Fig. S1 of the revised supplementary material). Therefore, except for the systematic and random uncertainties that we consider for the observations and model simulations, there will be no other missing factors which could rival to the emission rates.

We agree that the uncertainties of observations and models are important and will clarify this point in the revised manuscript:

"Therefore the only factor that the model failed to replicate the observation, except for their uncertainties, was the emission trend." (Page 5, Lines 33-34)

And the methodologies and results of the estimation of systematic and random uncertainties in the observations and models will be clarified:

" From the average and a standard deviation of the monthly MAAP/COSMOS ratios, systematic and random uncertainties were estimated as ± 14% and ± 17%, respectively (± 22% in total)." (Page 4, Lines 8-9)

" We estimated the model uncertainties from meteorology to be ~± 16%, in simulating surface BC concentrations under conditions with negligible wet deposition, considering both horizontal and vertical inhomogeneities in the model and the spread of multi model simulations of CO over the East China Sea (Kong et al., 2019). The uncertainty was assumed to be contributed equally from systematic and random terms (± 12 %)." (Page 5, Lines 25-28)

Then, the uncertainties in the absolute emission correction factors (E(y)/REAS2.1(2008)) and their trends will be clearly mentioned as follows:

"Overall uncertainty in the estimated E(y)/REAS2.1(2008) values was estimated to be ± 27%, including those random and systematic from both model and observation (see Sect. 2). On the other hand, the uncertainty in its trend was estimated to be ± 21%, as influenced only by random uncertainties." (Page 9, Lines 9-11)

In Fig. 7a, the uncertainty range (a band with pale red color) will be expanded accordingly, to cover the overall uncertainty.

4. Figures 3 and 4. Why stress spring and select spring 2018 for comparing the modeling and observation results? Any special reasons?

We just selected this period as an example and did not have any particular intention with the selection. We will provide time series plots during the whole period (2009–2019) as Fig. S1 in the revised supplementary material.

6. Small issue. What are the meanings of the dots with two colors in Figure 10b?

In legend of the figure we will show that they are yearly and 3-y running mean values in the revised manuscript.

We again thank the reviewer for the important suggestions.

---

## Author Comment (AC3) · 26 Mar 2020

**Response to the Reviewer #3**

The manuscript uses 11-year measurements of BC at a single site downwind of China to estimate emission trends of BC in different regions of China. Meteorological variability was estimated based on the WRF-CMAQ model with fixed emissions. Although only one site is used, the 11-year-long data record has valuable information that can be used to constrain source emissions.

We appreciate the reviewer's positive comment on our manuscript. Detailed point-by-point responses are given below.

1) My main concern is on the robustness of the regional trends derived with this single site. The regional trends were estimated by sampling observations and WRF-CMAQ model outputs by the footprint of HYSPLIT trajectories (c.f. Figure 5). Neither WRFCMAQ model nor HYSPLIT footprint was evaluated with regards to their respective ability to simulate interannual variability of meteorology, particularly precipitation. What is the resolution of the GDAS1 meteorology to drive HYSPLIT? The manuscript does not specify the resolution; it might be as coarse as 1degree x 1 degree. Since the authors used two models, they should be cross-validated. For validation of the HYSPLIT footprint, the footprint for one region should be compared to WRF-CMAQ simulation with that region's emissions turned off to validate if the two methods provide consistent variability of BC at the receptor site.

The horizontal resolution of the GDAS1 meteorology to drive HYSPLIT backward trajectories was $1 \times 1$ degree. This information will be included in the revised manuscript. The resolution is similar to that used in the WRF/CMAQ model (ca. 80 km). As the NCEP Final Analysis data (ds083.2), on which the WRF model is based, originated from GDAS, high compatibility between the HYSPLIT backward trajectories and WRF/CMAQ was expected by nature.

A quantitative discussion on the influence from precipitation on our analysis will be included as follows, using the $\Delta BC/\Delta CO$ ratio (Kanaya et al., 2016), for both observations and model simulations, to conclude that the influence is negligible, due to the adopted criteria (APT < 1 mm):
" The median $\Delta BC/\Delta CO$ ratio for the observational data with APT < 1 mm ($N = 26423$) was only 1.9 % lower than that for data with APT = 0 mm ($N = 18907$), suggesting insignificant influence." (Page 4, Lines 33- Page 5, Line 2, in the track change document)
" Wet deposition was represented with the cloud_acm_ae5 module. Similarly to the observational data, the modeled $\Delta BC/\Delta CO$ ratio decreased with APT; the modeled median $\Delta BC/\Delta CO$ ratio for data with APT < 1 mm ($N = 26737$) was 3.7 % lower than that for data with APT = 0 mm ($N = 19197$). The removal in the model appeared stronger than the observational trend (1.9 %) but the error introduced due to the wet deposition representation was estimated to be small (−2 %) when using the adopted criteria (APT < 1 mm)." (Page 5, Lines 14-18)

Thus the model's reproducibility of the wind field and its interannual variability is of main concern. Figure S3 (as follows, to be included in the revised supplementary material) compares the observed and modeled (i.e., WRF) monthly-averaged zonal and meridional wind speeds during the whole study period at 925 hPa, over 5 meteorological observatories surrounding the Fukue site, i.e., over Qingdao, Shanghai, Fukuoka, Kagoshima, and Gosan. The absolute wind speeds and their month-to-month and interannual variations agreed quite well.

Nonetheless the origin region assignment will have uncertainty due to incompleteness of the used meteorological field. In the revised manuscript, uncertainty in the region assignment will be evaluated against trajectories based on the ERA5 meteorological field with a finer horizontal resolution (30 km) during 2013−2015, and the following statement will be included:

"While the assignment agreed for large fractions (83−93 %) on a country level (i.e., China Korea, and Japan), the successful fraction decreased to 57−67 % for the four individual Chinese regions, due to crosstalk with the adjacent regions." (Page 4, Lines 25-29)

[Figure]

[Figure]

**Figure S3.** Observed monthly-averaged zonal (u) and meridional (v) wind speeds at 925 hPa at 00Z over five locations near Fukue (grey lines) were compared with those at corresponding girds of WRF/CMAQ model simulations.

2) For the validation of WRF-CMAQ, although the manuscript did compare WRF-CMAQ model against the EMeRGe-Asia campaign, that comparison was only for one spring season (2018) while the main use of the model was to estimate interannual variations of BC. It is the WRF-CMAQ model's ability to simulate interannual variability of meteorology that should be evaluated. For example, the manuscript should present WRFCMAQ simulated meteorological parameters at Fukue Island with those observed, and such comparison should be done for interannual scale. If data are available, the WRFCMAQ model's meteorology over mainland China should be evaluated too.

We will include the time-series plots comparing BC and CO observations with the WRF-CMAQ results for the whole study period (2009–2019) as Figure S1 in the revised supplementary material. There we demonstrate that individual peaks were well captured by the model not only during spring 2018 but also during the whole period. The reproducibility of the interannual variability of meteorology (wind field) was verified (shown above) and will be demonstrated as Figure S3 of the revised supplementary material.

We thank the reviewer again for the productive comments.

**References**

Kanaya, Y., Pan, X., Miyakawa, T., Komazaki, Y., Taketani, F., Uno, I., and Kondo, Y.: Long-term observations of black carbon mass concentrations at Fukue Island, western Japan, during 2009–2015: constraining wet removal rates and emission strengths from East Asia, Atmos. Chem. Phys., 16, 10689–10705, https://doi.org/10.5194/acp-16-10689-2016, 2016.

---

## Author Response (AR2)

**Reply to Referee #2:**

I agree with the authors that REAS might be quite different from the "true" emission flux, then why did you choose this inventory? The authors need to explain the reasons of emission data selection in Method section.

Reply. Practically our model analysis using REAS2.1 started in 2011 and were extended several times to cover the whole period. In the early days the REAS2.1 was one of the up-to-date emission inventory data sets. In text, it is more important to conclude that selection of emission inventory is not critical for the major results of this study about the emission trend estimation. In the revised manuscript, we included the following sentences at the end of Section 2:

The bias in the emission inventory from true values is most likely systematic, i.e., associated with systematic uncertainties in emission factors and/or activities common over a country or its individual sub region. Therefore we aim for correction of the systematic bias with the E(y)/REAS2.1(2008) ratio in this study. The geographical distribution pattern of BC emission (Fig. 1) needs to be less uncertain; this condition is satisfied with REAS2.1, even though it is a relatively old emission inventory. This is inferred from the fact that the simulated BC mass concentrations using the emission inventory have more systematic bias than random discrepancies from observations, as shown in Sect. 3.2. Indeed, the geographical pattern of BC emission is almost unchanged with more recent emission inventories. Therefore we conclude that selection of emission inventory is not critical for the major results of this study about the emission trend estimation.

We again thank the referee #2 pointing to the critical part of discussion and improving the manuscript.

**List of all relevant changes made in the manuscript**

1. Page 6, Lines 2-8. The following sentences were inserted upon a comment from Referee #2.

[revised manuscript text omitted]